# Quantum game strategy solution for R&D cartel: Reorganizing government R&D investment strategy in Korea

**Dongkyu Won**[ID]**, Jongyeon Lim**[ID]**, BangRae Lee**[ID]*

Center for R&D Investment and Strategy Research, Korea Institute of Science and Technology Information, Seoul, Republic of Korea

* brlee@kisti.re.kr

**Data Availability Statement:** All relevant data are within the paper and its Supporting information files.

**Funding:** Development and Utilization of Innovation Strategy Analysis Models for the Science and Technology Industry (K-23-L03-C04). The funders

## Abstract

This study reevaluates criticism of the Korean government's R&D investment strategy, which was considered an 'R&D cartel' and was cited as a reason for the Korean government's R&D cuts in 2023, through an advanced framework of quantum game theory. By modeling the interaction between the government and researchers as a quantum information strategy game, it redefines the dynamics of R&D investment as a quantum game involving two key players: the R&D manager (Alice) and the research performer (Bob). This quantum game, akin to the Prisoner's Dilemma but focused on responsibility and sincerity, allows for the exploration of strategic complexities and decision-making dynamics not possible in classical models. It introduces quantum entanglement and superposition as innovative strategies to shift the paradigm of R&D investment, suggesting that terms like 'R&D bureaucracy' and 'R&D monopoly' more accurately describe the moral hazards in this sector than 'R&D cartel'. Through simulations, the paper demonstrates how quantum strategies can significantly alter outcomes, providing fresh insights and policy alternatives for R&D innovation. This research not only challenges conventional investment frameworks but also proposes a novel approach for achieving Pareto optimal outcomes in government R&D investments, emphasizing the transformative potential of quantum game theory in strategic decision-making and policy development.

## Research background and purpose

The Korean government, acknowledging the inefficiencies in the management of its R&D budget, has recalibrated its financial allocation for R&D in 2024. The revised budget, set at 25.9 trillion won, marks a 16.6% reduction from the previous year [1]. This decision was ratified at the plenary session of the National Assembly, ultimately confirming the budget at the stated amount. Notably, this reduction represents a 15% decrease from the 31.1 trillion won allocated in 2023. Consequently, the investment proportion of the Korean government's total expenditure has diminished from 4.9% to 3.9%. This shift marks a significant moment in the nation's fiscal history, as it is the first time since 1991, a span of 33 years, that the proportion of

had no role in study design, data collection and analysis, decision to publish, or preparation of the manuscript.

**Competing interests:** The authors have declared that no competing interests exist.

investment relative to total government expenditure has plummeted to the 3% range, a phenomenon last observed 21 years ago [2]. Sources such as "South Korea, a science spending champion, proposes cutbacks" [3] and "South Korean scientists' outcry over planned R&D budget cuts" [4] offer detailed insights into these developments.

The Korean government rationalized this decision, citing 'insufficient visible results' in the R&D sector, 'a proliferation of small-scale R&D projects in the form of sharing', and 'a closed research system centered on domestic researchers and funds' [2]. Within this context, the term 'R&D cartel', diverging from its traditional usage, emerged as a critical descriptor for the inefficient management and allocation of R&D funds. Unlike a conventional cartel that typically implies coordination of price and production among competitors, the 'R&D cartel' in this context refers to a group or project monopolizing resources or influencing the decision-making process [5]. In this scenario, terms such as 'R&D bureaucracy' or 'R&D monopoly' may more accurately encapsulate the situation.

This study posits that the employment of these terms, in relation to Korea's R&D budget cuts, functioned more as rhetorical devices to justify policy shifts than as reflections of economic or legal precision. The study adopts a game theory perspective to analyze the cause of these developments, aiming to identify viable policy alternatives. From this angle, depending on the perspective from which the cartel is viewed, it can influence market competition and innovation. Therefore, an academic analysis is warranted over a purely political interpretation.

The objective of this study is to apply quantum game theory to offer novel strategic insights and interpret the complex dynamics inherent in R&D investment decisions. Quantum game theory expands upon traditional game theory, exploring new strategic possibilities through the unique characteristics of quantum entanglement and superposition. This approach facilitates the exploration of strategic scenarios that classical game theory cannot adequately address. Consequently, this study aims to contribute to the understanding of the prisoner's dilemma in R&D investment decision-making from the perspective of quantum game theory, thereby presenting pertinent issues and alternatives related to R&D investment strategies.

## The quantum game model

### Development of game theory: Classical and quantum perspectives

Game theory offers a mathematical framework for analyzing interdependent decision-making situations in a logical manner. This theory is utilized to elucidate a myriad of scenarios where a player's payoff is contingent not only upon their own actions but also on the actions of others. Quantum mechanics shares similarities with the stochastic model, which delineates the state of particles and their transformations via a wave function. This model is predicated on the probability of detecting the physical quantity under observation.

The foundational aspects of classical game theory were established by Von Neumann and Morgenstern in 1953 [6]. John Nash's proposition of the Nash equilibrium in 1996 furthered this theoretical framework [6]. In 1999, Meyer proposed a quantum game concept, and Eisert et al. [7] introduced the quantum prisoner's dilemma game, presenting a methodology for participants to surmount classical dilemmas.

The concepts of superposition, entanglement, and decoherence within quantum mechanics broaden the strategic landscape in decision-making processes. They introduce a dimension that stands in contrast to the principles of reality, causality, and locality prevalent in classical mechanics. Traditional prisoner's dilemma games are generally constrained to binary options (A or B). However, quantum mechanics facilitates strategic alterations through the superposition of A and B. These quantum game theories have gained widespread application in various domains, including economics, social sciences, communication, and biology, where they have

significantly contributed to the evolution of logic [8]. Classical game simulation theory uses a deterministic approach focusing on rational choices and optimal strategies of players. It concentrates on finding equilibria in games, which means identifying the best strategy for each player in various interactive situations. A typical example is the Nash equilibrium.

On the other hand, quantum game simulation theory applies principles of quantum mechanics to game theory, proposing a new type of game model. It explores new strategic spaces impossible in traditional game theory by utilizing quantum mechanical phenomena like superposition and entanglement. This approach deals with game outcomes probabilistically, with player choices leading to probabilistic rather than deterministic outcomes.

The distinction between classical and quantum game simulation theories marks a significant evolution in the field of game theory, broadening the scope of strategic analysis and application in uncertain environments.

In particular, the field of integrating AI and Markov models, etc. with classical and quantum game theories is an active area of research, and while specific titles may vary, the suggested papers indicate the focus and approach researchers might take in these interdisciplinary studies.

In the case of classical game theory, various studies combining Markov models and stochastic strategy models [9–13] and models incorporating AI models [14–18]. Meanwhile, in the case of quantum game models, various studies combining Markov models and stochastic strategy models and models incorporating AI models [19–23]. Whether classical game theory or quantum game theory, the convergence of Markov models, stochastic strategy models, and AI models complements each other's weaknesses, enabling more accurate predictions. Comparing the characteristics of convergence models for classical and quantum game theory using Markov and stochastic strategy models and AI models can be conceptualized as shown in Table 1.

This paper aims to explore the potential of quantum strategy games to transcend the classical prisoner's dilemma, offering new insights into overcoming traditional strategic limitations. The interplay of quantum mechanics and game theory suggests a paradigm shift in

**Table 1. Comparison of characteristics of convergence application models by game theory.**

| Game Theory Type | Model Type | Convergence Model Characteristics |
|---|---|---|
| Classical Game Theory | Markov Model Stochastic Model | Integrates time-based transitions with rational decision-making, allowing for dynamic strategy evolution in predictable environments. |
| | AI Model | Uses historical data and pattern recognition to refine strategies and predictions within classical game frameworks. |
| Quantum Game Theory | Markov Model Stochastic Model | Adapts quantum probability and dynamics to a temporal context, potentially capturing the evolution of entangled strategic choices over time. |
| | AI Model | Employs quantum machine learning algorithms that might take advantage of quantum computational speedups to optimize strategies in complex game scenarios. |

This table compares classical and quantum game theory models, where classical models use rational decision-making and historical data for strategy development, while quantum models incorporate quantum mechanics principles, allowing for entangled decision dynamics and potentially leveraging quantum computational advantages for strategy optimization.

understanding strategic decision-making processes, particularly in scenarios traditionally modeled by classical game theory.

The Schrödinger equation, a foundational equation in quantum mechanics, plays a crucial role in the quantum game theory, particularly in understanding the quantum game version of the Prisoner's Dilemma. Understanding this equation is essential for grasping the quantum game theory aspect of the Prisoner's Dilemma, which is based on an evolutionary model incorporating concepts central to quantum mechanics such as wave functions, operators, and eigenstates.

The Schrödinger equation describes the time evolution of a wave function, which represents the state of a quantum system [24]. Eigenstates represent the system's fundamental states, where the wave function remains unchanged by the application of an operator. This phenomenon, where a special case becomes the norm in the quantum world, links directly to the reversal of ordinary and special cases in quantum mechanics compared to classical mechanics.

In the quantum game version of the Prisoner's Dilemma, players' strategies are represented by wave functions instead of the pure or mixed strategies found in traditional game theory. The game's goal is to find the optimal solution through the superposition of quantum strategies. The eigenstate of the wave function plays a crucial role in this process. Finding an eigenstate in quantum mechanics, akin to finding the essential state of an electron, parallels the process of players finding their optimal strategies in the Prisoner's Dilemma quantum game.

The evolutionary model in quantum game theory applies these quantum mechanical concepts to game theory, exploring how players' strategies can evolve and optimize over time. Basic principles of quantum mechanics, such as those in the Schrödinger equation, deeply influence the rules and outcomes of the game. Consequently, the quantum game version of the Prisoner's Dilemma surpasses traditional game theory, enabling the discovery and understanding of new strategies.

Simulations of this model include applying various forms of wave functions to operators and observing the resulting changes to identify eigenstates. These simulations are crucial for understanding the evolutionary development of strategies in quantum game theory, ultimately illustrating how participants can achieve optimal outcomes through cooperation and competition.

## Classic R&D investment game model

This study explores a game-theoretic model of R&D investment strategy, focusing on two key participants: a government R&D budget efficiency manager (Alice) and a research project executor (Bob). The game is designed as a sequential or dynamic game, wherein the government R&D budget efficiency manager's types are categorized based on their tendency towards efficient ($A_a$) or inefficient ($A_b$) budget investment. Similarly, research project executors are distinguished by their orientation towards stable ($B_a$) or unstable ($B_b$) project execution.

Alice is responsible for managing the efficient use of the R&D budget, aiming to select projects that promise the highest possible investment returns. The ideal scenario involves using the budget efficiently to select and successfully implement projects with high return on investment. However, projects with high investment potential often carry a significant risk of failure, presenting a liability risk for managers. Conversely, projects with lower risk typically offer lesser investment returns, creating a strategic dilemma.

On the other hand, Bob is responsible for the success or failure of research projects, typically opting for tasks with a higher probability of success [25]. This approach often aligns with lower incentives, whereas tasks with higher risk levels offer greater rewards.

In this context, the study defines and analyzes the potential strategic combinations between Alice and Bob, exploring the dynamics of cooperation and defection between them. The interactions and scenarios that may arise in this R&D investment strategy game between Alice and Bob are influenced by the efficiency of budget investment and the stability of project execution, providing important insights for understanding strategic decision-making in the management and execute on of R&D processes.

In this context, the study defines and analyzes the possible strategic combinations between Alice and Bob, exploring the dynamics of cooperation and defection bet.

ween them as follows:

- Alice ($A_a$) and Bob ($B_a$): Alice efficiently manages the R&D budget, while Bob conducts research with a focus on stable project execution. This combination leans towards high-risk projects, offering significant rewards upon success, but with shared accountability in case of failure.

- Alice ($A_a$) and Bob ($B_b$): Alice efficiently manages the budget, but Bob undertakes research with unstable execution. Here, Alice may select ambitious projects (large/long-term projects), but Bob's unstable approach increases the risk of project failure.

- Alice ($A_b$) and Bob ($B_a$): Alice adopts a less efficient approach to budget management, while Bob remains focused on stable project execution. In this scenario, Alice tends to choose lower-risk projects (small/short-term tasks), resulting in low risk but also limited rewards.

- Alice ($A_b$) and Bob ($B_b$): Both parties exhibit tendencies towards inefficient budget management and unstable project execution. They are likely to opt for the safest projects (small/short-term tasks), which involve minimal risk and reward. In case of failure, the probability of accountability issues arising between them is high.

The study presents a quadrant model to illustrate the potential strategic combinations between Alice (efficient or inefficient R&D budget manager) and Bob (stable or unstable research project executor). Alice's strategies are divided into efficient or inefficient budget management, while Bob's approaches are categorized as stable or unstable project execution. Each cell within the quadrant delineates a distinct strategic combination for Alice and Bob, offering a structured framework to analyze the interaction dynamics in the R&D investment strategy game.

In Table 2, various scenarios are depicted based on the strategic combinations of Alice and Bob. These scenarios highlight potential interactions and outcomes:

- Alice ($A_a$) and Bob ($B_a$): This scenario portrays a situation where Alice efficiently manages the R&D budget and Bob conducts research with a focus on stable project execution. The anticipated results are high performance in large/long-term tasks, culminating in success.

- Alice ($A_a$) and Bob ($B_b$): Here, Alice manages the budget efficiently, but Bob adopts an unstable approach to research execution. From Alice's perspective, this combination yields the least favorable return on investment. However, for Bob, it still secures a medium level of success, especially in large-scale/long-term tasks.

- Alice ($A_b$) and Bob ($B_a$): In this instance, Alice opts for smaller/shorter-term tasks to minimize budgetary inefficiency, while Bob remains focused on stable project execution. The outcome is moderately satisfactory from Alice's standpoint. However, for Bob, it is viewed as a relative failure due to the mismatch between the effort exerted and the incentives received.

**Table 2. Classic game framework of R&D investment strategy.**

| Bob / Alice | $(C_b)$ Unstable Project Execution | $(D_b)$ Stable Project Execution |
|---|---|---|
| $(C_a)$ Efficient Budget Investment | High performance, success R, R (3, 3) | Worst performance, medium success S, T (0, 5) |
| $(D_a)$ Inefficient Budget Investment | Medium performance, failure T, S (5, 0) | Low performance, limited success P, P (1, 1) |

This table portrays a classic game framework for R&D investment strategy, depicting outcomes for pairs of government managers (Alice) and researchers (Bob) based on efficient or inefficient budget investment and stable or unstable project execution, with payoffs reflecting various degrees of success and failure.

• R (Reward): Reward when two players cooperate.

• S (Sucker): Reward when only one player cooperates and the other player defects.

• T (Temptation): Reward when one player defects and the other cooperates.

• P (Punishment): Reward when both players defect.

• Alice ($A_b$) and Bob ($B_b$): Both Alice and Bob demonstrate a lack of commitment in this scenario. Alice opts for inefficient budget management, and Bob's approach to research execution is unstable. Consequently, overall performance deteriorates, and success in tasks is limited.

Analyzing the R&D investment game as described above, to maximize each player's utility, Alice can focus on short-term/small-scale investments ($A_b$) to minimize risk, while Bob can also focus on short-term/small-scale tasks ($B_b$) to optimize his utility. This interaction results in a Nash equilibrium [26], a state where neither party has an incentive to deviate from their current strategy, assuming the other maintains their strategy as well. This scenario is indicative of 'moral hazard' [27]. The remuneration structure in this context is outlined as follows.

Table 2 presents the classic game payoff matrix for the R&D investment strategy game, reflecting a specific payoff structure. This table enables the analysis of game equilibrium within a scenario that mirrors a Prisoner's Dilemma situation.

• T(5) > R(3): This inequality suggests that for researchers (Bob), pursuing short-term, unstable projects (high-risk, high-reward scenarios) is more advantageous than engaging in stable, long-term research. It indicates a preference for immediate gains over sustained, stable project execution.

• R(3) > P(1): Investing in efficient budget management and undertaking stable research projects yield better results compared to a scenario where both parties opt for inefficient budget management and unstable project execution. This implies that the optimal collaboration between Alice's efficient budget investment and Bob's stable project execution leads to better outcomes.

• R(3) > (T(5) + S(0))/2: The total reward from both parties choosing their optimal strategies (efficient budget management and stable project execution) exceeds the reward from any other combination of strategies.

In this game scenario, each player, Alice and Bob, selects a strategy that maximizes their individual benefit. However, this approach often results in a reduced overall gain, manifesting as a cycle of 'low-risk, low-reward' characterized by both inefficient budget management and

unstable project execution. Alice, representing budget management, tends to adopt a 'play it safe' strategy aiming to minimize risk and personal liability. Conversely, Bob as a researcher prefers immediate rewards, even in the context of long-term investments, which diminishes the potential for undertaking large-scale, stable projects.

The result of these interactions converges to an equilibrium characterized by 'low investment, low performance,' akin to the Nash equilibrium found in the prisoner's dilemma. In these scenarios, each player independently makes choices that are optimal from their perspective, while the collective outcome is suboptimal [28]. This situation is elucidated through a detailed analysis of the classic Prisoner's Dilemma game, enhancing our understanding of the dynamics at play.

## Quantum information strategy game model composition

This study aims to establish a theoretical foundation centered on the "EWL (Eisert, Wilkens, and Lewenstein) quantization protocol [7]." A critical aspect of transitioning from classical to quantum game theory is the application of ring homomorphism. In algebra, ring homomorphism is defined as a function that preserves specific operations (usually addition and multiplication) between two algebraic structures, known as rings. For instance, consider a function f: $R \rightarrow S$ between any two rings R and S. This function f must satisfy certain conditions to be considered a ring homomorphism [29]:

For addition conservation:

$$f(x + y) = f(x) + f(y) \text{ for any elements x, y in R} \tag{1}$$

For multiplication conservation:

$$f(xy) = f(x)f(y) \text{ for any elements x, y in R.} \tag{2}$$

These quasi-isomorphisms are instrumental in the mapping process that translates strategic choices from classical game theory into states and operators in quantum game theory. In this context, the strategies of 'cooperate' and 'defeat' in classical game theory correspond to the quantum states $|0\rangle$ and $|1\rangle$, respectively. In quantum mechanics, these vectors are referred to as eigenstates. Hence, the eigenstate of $\sigma x$ comprises a superposition of $|\text{Defection}\rangle$ and $|\text{Cooperation}\rangle$ states. Consequently, there are four quantum-classical correspondences:

$$|00\rangle \longleftrightarrow (C, C), \tag{3}$$

$$|01\rangle \longleftrightarrow (C, D), \tag{4}$$

$$|10\rangle \longleftrightarrow (D, C), \tag{5}$$

$$|11\rangle \longleftrightarrow (D, D). \tag{6}$$

Introducing a semi-isomorphic function $\rho$ transforms the strategies of classical game theory into quantum game theory. This function is defined as $\rho(i_c) = I$ (identity matrix) and $\rho(r_c) = \sigma_x$ (Pauli X matrix). This mapping forges a link between classical and quantum game theories, where these states in quantum mechanics are regarded as distinct states. This conceptual framework thus allows for an enriched understanding of strategic interactions in the realm of quantum game theory.

In this study, quantum game theory is employed in conjunction with the concept of neuro-cognitive modeling, as referenced from the work of Surov [30]. Players engaged in a game within an R&D system framework can be conceptualized and analyzed in a manner similar to a

**Fig 1. Concept map for quantum information strategy game.** In the diagram, a 'Black Box' represents the R&D system, and the cognitive attributes of the players are depicted as overlapping circles, labeled with 'Cognition' and 'Mind' for players named Alice and Bob, symbolizing their mental states and cognitive processes. The 'Observation' section symbolizes the transition of these cognitive processes into actual behaviors or actions. The 'Potential State' reflects the strategic options available to the players, corresponding to decisions like 'Cooperation' and 'Defection.' The probabilities of these strategies are represented in a manner akin to the squared magnitude of probability amplitudes in quantum mechanics, which calculates the likelihood of each strategy.

system device in quantum physics (Fig 1). This approach allows the cognitive attributes and decision-making processes of the players to be modeled as quantum mechanical properties, analyzing how various potential actions and decisions within the game interact with each other.

In this context, the rotation equation used in quantum game theory represents the possible action states of the players, mathematically expressing their strategic choices. Each state in this equation is associated with a specific strategy or decision, and the probabilistic distribution of these strategies can be interpreted as wave function probability amplitudes in quantum mechanics. In other words, the probability of each strategy is proportional to the square of the coefficients in the equation.

As a result, this study provides a novel methodology for analyzing and predicting human cognitive and behavioral patterns. The psychological states and potential actions of game participants are expressed in the mathematical language of quantum mechanics, allowing for a more sophisticated modeling of their complex interactions.

The study proposes five distinct quantum strategies as fundamental elements of quantum game theory. These strategies are denoted as I (Identity matrix), H (Hadamard matrix), $\sigma_x$ (Pauli X matrix), $\sigma_y$ (Pauli Y matrix), and $\sigma_z$ (Pauli Z matrix). The overarching framework of the quantum information strategy game model, which is elaborated later in this paper, is depicted in Fig 2. This model emphasizes the determination of the game's equilibrium value through the quantum information computation of the game's final state. The procedural framework for the quantum game encompasses the following steps:

- Setting of Initial State: Establishing the initial conditions or state of the game.

- Entanglement Operator Setting: In quantum mechanics, an operator is employed to transform an existing quantum state into a new one. The entanglement operator is a critical component in this process.

- Selection of Strategy: Players choose their respective strategies, which are represented by quantum operations.

- Resolution of Entanglement Operators: This step involves the application and subsequent resolution of the entanglement operators, leading to a change in the quantum state of the system.

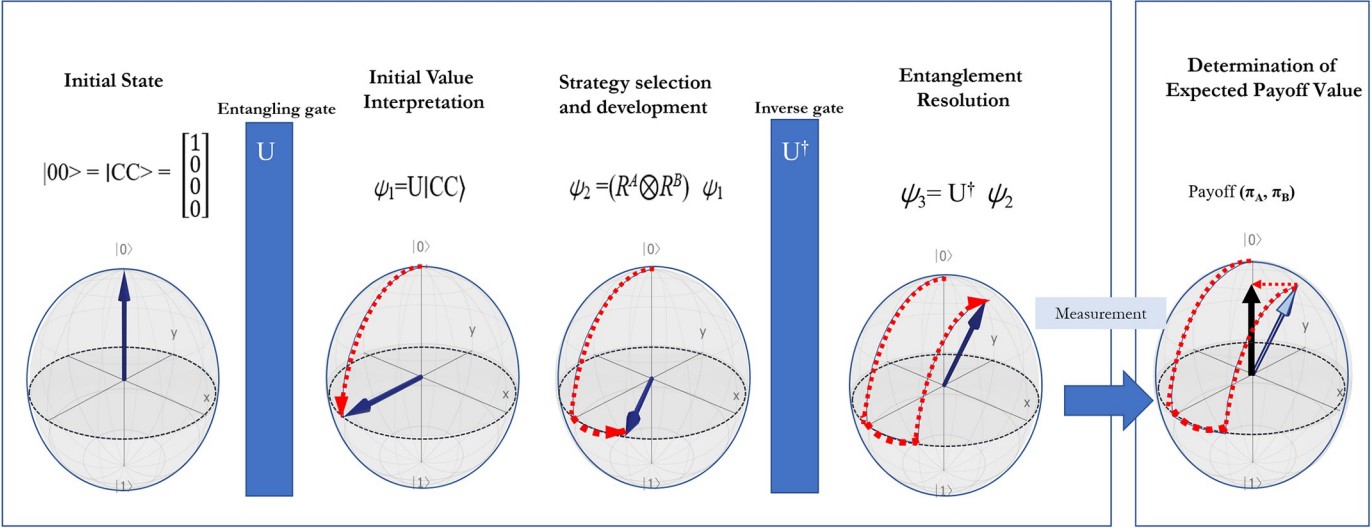

**Fig 2. Quantum information strategy game model.** Fig 2. illustrates the stages of a quantum information strategy game model: starting from a cooperative initial state, moving through an entanglement phase to introduce correlations, followed by strategy selection and development. It then progresses to disentangle and resolve the state, culminating in the measurement of outcomes to determine the players' expected payoffs. Initial State: Both players start in a cooperative state $|00\rangle$, represented on a Bloch sphere. Entangling Gate (U): An entangling operation introduces quantum correlations between the players. Initial Value Interpretation: The entangled state represents a mix of cooperative and competitive strategies. Strategy Selection ($R^A$ and $R^B$): Players choose strategies affecting the game's evolution. Entanglement Resolution ($U^\dagger$): The entangled state is resolved, setting the stage for payoff determination. Determination of Expected Payoff: The game concludes with a measurement that determines the players' payoffs, represented by $\pi A$ and $\pi B$.

- Confirmation of Final State by Observation: The final state of the game is determined through observation, post the application of strategies and resolution of entanglement operators.

This methodology integrates quantum mechanics principles with game theory, offering a unique perspective in understanding and analyzing strategic interactions within the R&D system. By applying quantum strategies and entanglement principles, this approach provides a novel framework for exploring decision-making processes in complex systems.

These procedures are pivotal in integrating fundamental quantum properties, such as superposition and entanglement, into quantum games. Entanglement, a key concept in quantum game theory, involves the interconnection of two quantum states, such that the measurement of one state instantaneously influences the other. This aspect is crucial for delving into quantum mechanics, particularly for understanding the role of entanglement in the quantum adaptation of the Prisoner's Dilemma game.

A significant advancement in quantum game theory is its ability to introduce a new tier of strategic complexity and potential. It achieves this by expanding upon the deterministic strategies of traditional game theory, extending them into the realm of quantum mechanical states. In quantum games, the central concepts of quantum entanglement and superposition facilitate novel strategic scenarios that were unattainable in conventional game theory. For instance, quantum entanglement encompasses the phenomenon where the strategies of two players directly impact each other, a dynamic absent in traditional game theory.

A crucial concept in the quantization process of game theory is the transformation of classical strategies into quantum game dynamics through semi-isomorphism. This transformation process maps strategic choices from the classical game into states within the quantum game.

Such a mapping significantly enhances the strategic complexity and diversity of the game, presenting a broader spectrum of strategic possibilities [31].

## Simulation results

### Quantum information strategy simulation analysis

In the quantum information strategy game, the strategies represented by quantum information vectors $|00\rangle$, $|01\rangle$, $|10\rangle$, $|11\rangle$, and so forth, exist within the Hilbert space. These vectors denote the respective quantum strategies (qubits) of the government and researchers. The initial part of the vector represents the government's strategy, while the latter part signifies the researcher's strategy.

From the perspective of the Hamiltonian, the rearrangement of any given equation in this context must adhere to the logical sequence of quantum state evolution within a quantum game or decision-making scenario. The analysis phase, as delineated by David McMahon [32], proceeds in the following order: state initialization, entanglement application, strategy selection, and entanglement resolution.

**Initial state.** The initial state of the system, denoted as $\psi_0$, must be established. This state reflects the players' initial strategies or positions before any game interaction (such as entanglement or strategic move) takes place. Although this initial stage might not be explicitly mentioned in the equations, it constitutes the implicit starting point within the Hamiltonian framework. Concerning the interpretation of the initial value $|00\rangle$, this vector indicates that the R&D manager (Alice) has chosen C (Cooperate) as the initial strategy. Similarly, the researcher (Bob)'s strategy is also represented by C (Cooperate).

**Initial value interpretation.** This representation implies that both the government (Alice) and the researcher (Bob) start the game with a cooperative strategy. This initial condition sets the stage for further strategic developments as the game progresses, with subsequent phases involving entanglement and strategic decision-making impacting the evolution of the game's state. The Hamiltonian approach to analyzing these quantum games provides a structured framework for understanding the complex dynamics of strategy selection and its implications in quantum decision-making scenarios.

$$|00\rangle = |CC\rangle = \begin{bmatrix} 1 \\ 0 \\ 0 \\ 0 \end{bmatrix} \tag{7}$$

In this study, an entangled state is created by applying an entanglement operator $U$ to the initial state. This procedure is crucial in quantum games, as it incorporates quintessential quantum properties such as superposition and entanglement into the system. Following the establishment of initial values, the next critical step is the generation of entanglement. Operators are applied to the initial state to engender a new system state. The newly formed system state, denoted as $\psi_1$, emerges as a superposition of states reflecting various decision combinations by the players.

The expression for $\psi_1$ is given by:

$$\psi_1 = U \mid CC\rangle \tag{8}$$

In this equation, $U$ represents the entanglement operator, instrumental in entangling the states of the players. The formulation of all entanglement operators is as follows, with the

entanglement angle preset to $\gamma = \frac{\pi}{2}$, $\Theta = \frac{\pi}{4}$, $\phi = \frac{\pi}{2}$:

$$U_{(\theta,Y)} = \frac{1}{\sqrt{2}} \left( I \otimes I + e^{iY} X \otimes X \right) \tag{9}$$

where X: $e^{\,i2\theta\sigma x} = \cos(2\theta)I + i\sin(2\theta)\sigma_x$
I: Identity matrix
$\sigma_x$: Pauli X matrix

Here, $\gamma$ is the entanglement parameter, which distinguishes it from the strategy in a classical game. When $\gamma = 0$, the strategy aligns with that of a classical game. In contrast, when $\gamma \neq 0$, entanglement occurs, signifying the commencement of a quantum game. Notably, $\gamma = \frac{\pi}{2}$ is indicative of maximum entanglement, with $\gamma$ ranging within $[0, \frac{\pi}{2}]$.

The entanglement operator $U_{(\Theta,\gamma)}$ symbolizes the mechanism through which the states of the two players are interconnected. The $X$ operator, a Pauli X matrix, is a unitary matrix representing a linear transformation. This transformation is critical as it preserves the normalization of the quantum state, maintaining both the length and angle (inner product structure) of the vectors. The application formula for the entanglement operator $U$ is structured as follows:

$$\psi_1 = U|CC> = \frac{1}{\sqrt{2}} \begin{bmatrix} 1 & 0 & 0 & -i \\ 0 & 1 & -i & 0 \\ 0 & -i & 1 & 0 \\ -i & 0 & 0 & 1 \end{bmatrix} \begin{bmatrix} 1 \\ 0 \\ 0 \\ 0 \end{bmatrix} = \frac{1}{\sqrt{2}} \begin{bmatrix} 1 \\ 0 \\ 0 \\ -i \end{bmatrix} \tag{10}$$

**Strategy selection and development.** When the system transitions into the entangled state $\psi_1$, players select their strategies, represented by rotation matrices $R^A$ and $R^B$. These strategic choices are pivotal in influencing the evolution of the system. The application of these strategies results in a new state, $\psi_2$, which is defined by $\psi_2 = (R^A \otimes R^B)\, \psi_1$. This transition corresponds to step 3 in the process and signifies the evolution of the state as a consequence of the players' strategic decisions [33].

$$\psi_2 = \left( R^A \otimes R^B \right) \psi_1 \tag{11}$$

Here, $\psi_2$ is contingent upon the strategies employed by the players. In a state of maximum entanglement, the chosen strategies of the players (which could be exemplified by a Pauli gate) have the potential to yield an optimal reward for both parties. Generally, $R^A$ and $R^B$ are quantum gates, determined by the strategies of players A and B, respectively. These quantum gates can be analogized to real-world or social scenarios, allowing for the application of approximate strategies. Within the Hilbert space of the fundamental state, each player manipulates their base state through personal beliefs and infers outcomes based on an N-dimensional vector space [34]. This study utilizes phase shifts and bit inversions in its analysis. Table 3 provides an example of how each gate can be interpreted as a strategic metaphor, illustrated through changes in bits and wavelengths.

The Hamiltonian is interpreted from an evolutionary standpoint, focusing on how various quantum gates, which represent strategic decisions in a quantum game, can influence the game's state. The analysis is structured as follows: Hamilton, which governs the evolution of the nation over time, is impacted by these gates, thereby dictating the unfolding of the player's strategy within the game. From a Hamiltonian perspective, these gates symbolize diverse strategic alterations to the player state in the quantum game [37].

**Table 3. Concept of quantum gate and strategy.**

| Gate \ Strategy | Conversion | Hamiltonian Perspective |
|---|---|---|
| *Identity Gate (I)* | This gate leaves the quantum state unchanged. $I : \alpha|0\rangle + \beta|1\rangle \Rightarrow \alpha|0\rangle + \beta|1\rangle$ | Analogous to maintaining the status quo in a game, without altering the current position or strategy |
| *Hadamard Gate (H)* | Introduces superposition, putting the qubit in a state where it is both 0 and 1 with equal probability. $H : \alpha|0\rangle + \beta|1\rangle \Rightarrow \alpha\frac{|0\rangle+|1\rangle}{\sqrt{2}} + \beta\frac{|0\rangle-|1\rangle}{\sqrt{2}}$ | Similar to a dynamic and exploratory approach in a game, where the player is open to different outcomes and adapts their strategy accordingly. |
| *Pauli X Gate (bit flip)* | Flips the qubit state from 0 to 1 and vice versa. $X : \alpha|0\rangle + \beta|1\rangle \Rightarrow \beta|0\rangle + \alpha|1\rangle$ | Analogous to a transformative approach that completely inverts the existing strategy. |
| *Pauli Y Gate (bit and phase flip)* | Combines a bit flip (like X) with a phase shift. $Y : \alpha|0\rangle + \beta|1\rangle \Rightarrow -\beta i|0\rangle + \alpha i|1\rangle$ | Represents a multifaceted approach encompassing both the player's position and perspective, similar to a radical and strategic shift in a game. |
| *Pauli Z Gate (phase change)* | Introduces a phase shift without changing the state itself. $Z : \alpha|0\rangle + \beta|1\rangle \Rightarrow \alpha|0\rangle - \beta|1\rangle$ | Analogous to altering the interpretation or methodology of a game without modifying the fundamental position. |

Table 3 delineates how different quantum gates influence the state of a qubit and compares these changes to strategic movements in game theory. The Identity Gate maintains the current state and the Hadamard Gate creates a state of superposition and exploration. The Pauli X Gate flips the state, representing a complete strategic overhaul and the Pauli Y Gate adds complexity with both bit and phase flips and the Pauli Z Gate alters the phase, signifying a change in approach without changing the actual strategy [35, 36].

Firstly, consider the scenario where $R^A$ = I and $R^B$ = I as an example. In this instance, the identity matrix strategy is utilized to simulate the effect of entanglement strength while maintaining the existing strategic direction unchanged.

$$\psi_2 = (I \otimes I)\,\psi_1 = \begin{bmatrix} 1 & 0 & 0 & 0 \\ 0 & 1 & 0 & 0 \\ 0 & 0 & 1 & 0 \\ 0 & 0 & 0 & 1 \end{bmatrix} \frac{1}{\sqrt{2}} \begin{bmatrix} 1 \\ 0 \\ 0 \\ -i \end{bmatrix} = \frac{1}{\sqrt{2}} \begin{bmatrix} 1 \\ 0 \\ 0 \\ -i \end{bmatrix} \tag{12}$$

The strategic maneuvers of the government and researchers are each linked to the evolution of the system through $(R^A \otimes R^B)\,\psi_1$. The state of each individual quantum strategy eventually transitions into the $\psi_2$ state following temporal evolution. For both the government and researchers, the single quantum information strategies $R^A_{(\Theta1, \phi1)}$ and $R^B_{(\Theta2, \phi2)}$ correspond to $\Theta_1, \Theta_2 \in [0, \pi]$ in phase and $\phi1, \phi2 \in [0, \frac{\pi}{2}]$ in amplitude.

**Entanglement resolution.** The final step involves resolving the entanglement using $U^\dagger$, which is the inverse function of the entanglement operator, thereby concluding the game.

$$\psi_3 = U^\dagger \psi_2 \tag{13}$$

Here, $\psi_2$ is shaped by the players' strategies. However, as elaborated earlier, if $U^\dagger$ is applied

to $\psi_2$ with $R^A = I$ and $R^B = I$, the result is as follows:

$$\psi_3 = U^\dagger \psi_2 = \frac{1}{\sqrt{2}} \begin{bmatrix} 1 & 0 & 0 & i \\ 0 & 1 & i & 0 \\ 0 & i & 1 & 0 \\ i & 0 & 0 & 1 \end{bmatrix} \frac{1}{\sqrt{2}} \begin{bmatrix} 1 \\ 0 \\ 0 \\ -i \end{bmatrix} = \begin{bmatrix} 1 \\ 0 \\ 0 \\ 0 \end{bmatrix} \tag{14}$$

Each player sends their quantum state to a final measuring device that determines their payoff. The final state of the quantum game is expressed as $P_{ee'} = |ee'|\psi_f\rangle|^2$. The lower quantum state element set is composed of the elements (CC, CD, DC, DD). As noted earlier, this quantum information strategy game reduces to a classical game when the entanglement factor $\gamma$ equals 0.

This model provides a nuanced understanding of the interplay between quantum strategies and their outcomes, offering insights into the dynamics of decision-making in quantum game theory. The transition from quantum to classical game scenarios through the manipulation of entanglement underscores the versatility and complexity of strategic choices in this framework.

**Determination of expected payoff value.** The determination of the expected payoff value involves confirming the final state through observation. The expected payoff for players A and B is derived based on the overlap of states. The square of the magnitude of each coefficient in the state vector $\psi_3$ represents the probability of a particular outcome. The final expected reward is calculated as a weighted sum of these probabilities. This approach is used to compute the expected payoffs of players A and B when $R^A = I$ and $R^B = I$.

For player A, the expected payoff $\langle \pi_A \rangle$ is calculated as:

$$\langle \pi_A \rangle = 3|\langle \psi_3 \mid CC|^2 + 0 \mid \langle \psi_3 \mid CD \rangle|^2 + 5 \mid \langle \psi_3 \mid DC \rangle|^2 + 1 \mid \langle \psi_3 \mid DD \rangle|^2 \tag{15}$$

$$\begin{aligned} &= 3|\langle \psi_3 \mid 1|^2 + 0 \mid \langle \psi_3 \mid 0 \rangle|^2 + 5 \mid \langle \psi_3 \mid 0 \rangle|^2 + 1 \mid \langle \psi_3 \mid 0 \rangle|^2 \\ &= 3 \end{aligned} \tag{16}$$

For player B, the expected payoff $\langle \pi_B \rangle$ is similarly calculated:

$$\langle \pi_B \rangle = 3|\langle \psi_3 \mid CC|^2 + 0 \mid \langle \psi_3 \mid CD \rangle|^2 + 5 \mid \langle \psi_3 \mid DC \rangle|^2 + 1 \mid \langle \psi_3 \mid DD \rangle|^2 \tag{17}$$

$$\begin{aligned} &= 3|\langle \psi_3 \mid 1|^2 + 0 \mid \langle \psi_3 \mid 0 \rangle|^2 + 5 \mid \langle \psi_3 \mid 0 \rangle|^2 + 1 \mid \langle \psi_3 \mid 0 \rangle|^2 \\ &= 3 \end{aligned} \tag{18}$$

The final expected payoff varies depending on the player's strategy and the specific state $\psi_2$. In the calculations shown in Table 4 and Fig 3, different matrices $R^A$ and $R^B$ corresponding to each strategy would yield different payoff values.

This methodology allows for a precise calculation of the expected payoffs in quantum games, considering the intricacies of quantum states and strategies. The ability to modify the payoff outcomes based on different strategic matrices underscores the dynamic nature of quantum game theory and its applicability in various decision-making scenarios.

## Application of player strategy

In this study, we aim to analyze changes in payoff values when $R^A$ and $R^B$ are configured as identity strategy, Hadamard strategy, and various Pauli operators. The simulation investigates the modifications in the compensation structure resulting from permutations of these

**Table 4. Payoff ($\pi_A$, $\pi_B$) according to combination of quantum strategies.**

| Combination of Strategies | $\pi_A$ | $\pi_B$ |
|---|---|---|
| ('I', 'I') | 3 | 3 |
| ('I', '$\sigma_x$') | 0 | 5 |
| ('$\sigma_x$', 'I') | 5 | 0 |
| ('$\sigma_x$', '$\sigma_x$') | 1 | 1 |
| ('I', 'H') | 0.5 | 3 |
| ('H', 'I') | 3 | 0.5 |
| ('H', '$\sigma_x$') | 3 | 0.5 |
| ('$\sigma_x$', 'H') | 0.5 | 3 |
| ('H', 'H') | 2.25 | 2.25 |
| ('$\sigma_x$', '$\sigma_y$') | 3 | 3 |
| ('$\sigma_y$', '$\sigma_x$') | 3 | 3 |
| ('$\sigma_y$', '$\sigma_y$') | 1 | 1 |
| ('$\sigma_x$', '$\sigma_z$') | 0 | 5 |
| ('$\sigma_z$', '$\sigma_x$') | 5 | 0 |
| ('$\sigma_z$', '$\sigma_z$') | 3 | 3 |
| ('$\sigma_y$', '$\sigma_z$') | 5 | 0 |
| ('$\sigma_z$', '$\sigma_y$') | 0 | 5 |

Table 4 presents the payoffs for players A and B based on various combinations of quantum strategies. The table highlights which strategies lead to Pareto optimal payoffs of 3 for both players, specifically when the strategies are ('I', 'I'), ('σx', 'σy'), ('σy', 'σx'), and ('σz', 'σz'), indicating situations where mutual cooperation yields the highest benefit.

quantum strategies, particularly under the influence of an entanglement degree set at $\frac{\pi}{4}$. To this end, the payoffs $\pi_A$ and $\pi_B$ were systematically analyzed under the following $R^A$ and $R^B$ strategy configurations:

- Both combined with the identity operator (I): ('I', 'I')

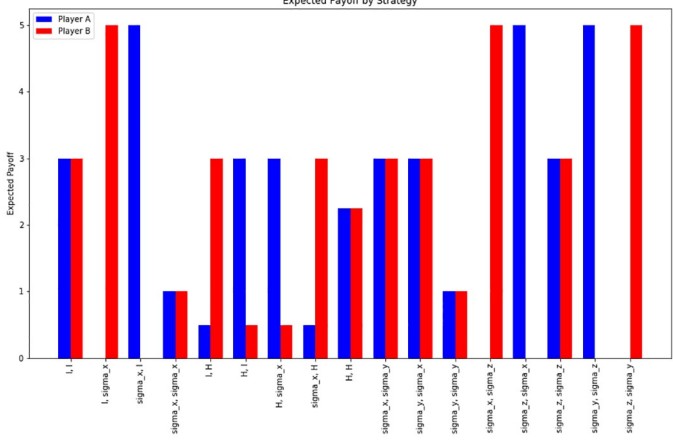

**Fig 3. Comparison of compensation systems according to combination of quantum strategies.**

- Combination of the I operator and the Pauli X operator in sequence: ('I', 'σ$_x$')

- Sequential combination of the Pauli X operator and the I operator: ('σ$_x$', 'I')

- Combining both with the Pauli X operator: ('σ$_x$', 'σ$_x$')

- Sequential combination of the I operator and the Hadamard (H) operator: ('I', 'H')

- Sequential combination of the Hadamard (H) operator and the I operator: ('H', 'I')

- Sequential combination of the Hadamard (H) operator and the Pauli X operator: ('H', 'σ$_x$')

- Sequential combination of the Pauli X operator and the Hadamard (H) operator: ('σ$_x$', 'H')

- Combining both with the Hadamard (H) operator: ('H', 'H')

- Sequential combination of the Pauli X operator and the Pauli Y operator: ('σ$_x$', 'σ$_y$')

- Sequential combination of the Pauli Y operator and the Pauli X operator: ('σ$_y$', 'σ$_x$')

- Both combined with the Pauli Y operator: ('σ$_y$', 'σ$_y$')

- Sequential combination of the Pauli X operator and the Pauli Z operator: ('σ$_x$', 'σ$_z$')

- Sequential combination of the Pauli Z operator and the Pauli X operator: ('σ$_z$', 'σ$_x$')

- Both combined with the Pauli Z operator: ('σ$_z$', 'σ$_z$')

- Sequential combination of the Pauli Y operator and the Pauli Z operator: ('σ$_y$', 'σ$_z$')

- Sequential combination of the Pauli Z operator and the Pauli Y operator: ('σ$_z$', 'σ$_y$')

For the aforementioned combinations of quantum strategies, both players reach the Pareto optimum of (3,3) with configurations such as ('I', 'I'), ('σ$_x$', 'σ$_y$'), ('σ$_y$', 'σx'), and ('σ$_z$', 'σ$_z$'), as detailed in Table 4. This analysis demonstrates the impact of various strategic configurations on the resultant payoffs, providing insights into the dynamics of quantum strategy applications in game theory scenarios. The quantum circuit diagram that implements this process is illustrated in Fig 4.

The diagram represents a two-qubit quantum circuit, which is used in quantum computing to perform operations on quantum bits (qubits). Here's what each part of the circuit does:

- Rz(π/4): This is a rotation gate that rotates the state of a qubit around the z-axis of the Bloch sphere by an angle of π/4 radians. It's applied to both qubits q0 and q1 at the beginning of the circuit.

- CNOT Gate: The plus sign with the circle and the line connecting it to another qubit represents a controlled NOT gate (CNOT). The qubit at the circle end is the control qubit, and

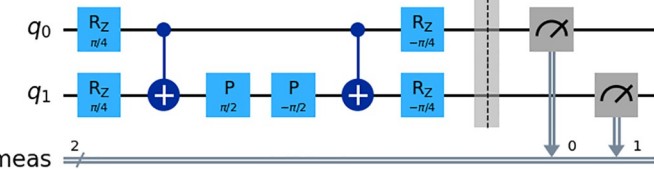

**Fig 4. Quantum circuit diagram when R$^A$ = X, R$^B$ = Y.** The image shows a quantum circuit for a two-qubit system, where specific operations are applied to simulate a game with strategies RA = X and RB = Y. The sequence includes rotations (Rz), controlled-NOT (CNOT) for entanglement, phase shifts (P), another set of rotations, and measurement operations, indicating the process of evolving the quantum state according to chosen strategies and then measuring the outcomes.

the qubit at the end of the line is the target qubit. If the control qubit is in the state $|1\rangle$, it flips the target qubit; otherwise, it does nothing.

- P($\pi$/2) and P(-$\pi$/2): These are phase gates that add a phase of $\pi$/2 and -$\pi$/2 to the qubit state, respectively. They only change the relative phase of the qubit state and not the probability amplitudes.

- Rz(-$\pi$/4): This is another rotation gate that rotates the state of a qubit around the z-axis by an angle of -$\pi$/4 radians.

- The vertical line with the double slash before the measurement indicates a barrier, which is used in quantum circuits to prevent certain optimizations that might otherwise be done by the quantum compiler. It's a way of telling the compiler to treat the operations before and after the barrier as distinct steps.

- The measurement gates (meas): At the end of the circuit, there are two measurement gates, indicated by the meter symbols. These gates measure the state of the qubits and collapse their state to either 0 or 1, which can then be read out as classical information.

- Regarding the $R^A = X$, $R^B = Y$ part, it might indicate the specific types of rotation to be applied on the qubits named RA and RB, with X and Y referring to rotations about the x-axis and y-axis of the Bloch sphere, respectively. However, this is not depicted in the circuit provided.

Meanwhile, in the second simulation, the combination ('$\sigma$x', '$\sigma$y') was selected from the Pareto optimal set as illustrated in Fig 4. In this scenario, the entanglement angle ($\gamma$) was varied from 0 to $\pi$ at intervals of $\frac{\pi}{32}$. The simulation aimed to observe the expected payoffs of players A and B ($\pi$A and $\pi$B) when adjusting the entanglement angle ($\gamma$) of the entanglement operator $U(\theta, \gamma) = \frac{1}{\sqrt{2}} (I \otimes I + e^{i\gamma} X \otimes X)$ over this range.

In Fig 5, player A employs the Pauli_X strategy, while player B utilizes the Pauli_Y strategy. This setup enables the observation of how the players' expected payoffs vary as the gamma value changes. Notably, the gamma value exhibits its maximum impact around $\frac{\pi}{4}$ (45 degrees).

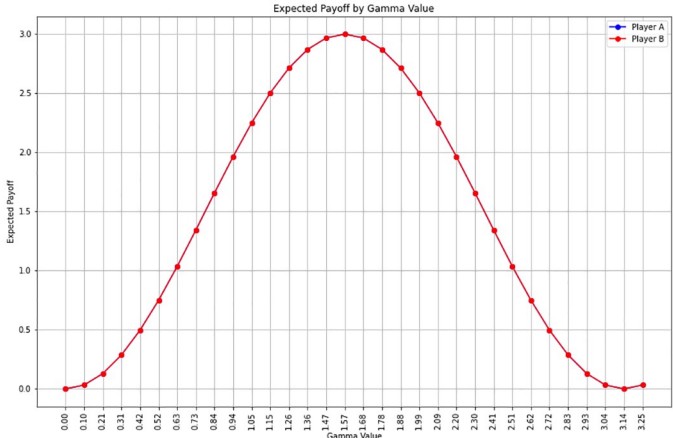

**Fig 5.** Change in entanglement angle ($\gamma$) and expected payoffs of players A and B. The graph displays the relationship between the entanglement angle (gamma value) and the expected payoffs for players A and B. It depicts a bell-shaped curve, with the payoff values on the y-axis and the gamma values on the x-axis. The peak of the curve suggests the optimal entanglement angle where the expected payoffs for both players are maximized. This figure illustrates how varying degrees of entanglement affect the strategic outcomes in a quantum game when players adopt specific strategies, in this case, Pauli X for player A and Pauli Y for player B.

This observation indicates that different combinations of quantum gates can yield varied outcomes, and the interplay between gate selection and the degree of entanglement crucially influences the final reward. Specifically, the entanglement angle at which the interaction is maximized, and consequently the maximum compensation value, was found to occur around $\frac{\pi}{4}$ (45 degrees).

This analysis underscores the significance of the entanglement angle in quantum game theory, particularly in terms of its effect on strategic outcomes. The results suggest that the optimal entanglement angle for maximizing rewards in this simulation context is approximately $\frac{\pi}{4}$, demonstrating the intricate relationship between quantum gate combinations and entanglement in determining player payoffs.

## Discussion

### Analysis of quantum game simulation results

The results obtained from quantum game simulations mark a significant advancement, transcending the traditional constraints of the classic Prisoner's Dilemma. Among the evaluated strategy combinations, four specific pairs of quantum strategies—('I', 'I'), ('σ$_x$', 'σ$_y$'), ('σ$_y$', 'σ$_x$'), and ('σ$_z$', 'σ$_z$')—achieved the Pareto optimal payoff of (3,3). This outcome illustrates a scenario where players can individually enhance their rewards without adversely affecting their counterparts, embodying the essence of an optimal cooperative strategy.

A particularly notable outcome is the achievement of Pareto optimality with the ('I', 'I') strategy. In the realm of static Hamiltonian quantum games, a static Hamiltonian implies that players' strategies or states remain constant throughout the game, analogous to actors adhering to a predetermined script. Their actions and outcomes are consistent and predictable, steered by the unalterable rules of static Hamiltonians.

This strategy, indicative of a static state, highlights the fact that Pareto optimality is attainable solely by manipulating the degree of entanglement. This discovery suggests that in practical applications, enhanced entanglement, akin to increased transparency and mutual trust, can lead to Pareto-optimal outcomes without necessitating substantial modifications to existing policy frameworks.

The implications of these simulation results are profound in the context of policy development. If maximal entanglement ($\theta = \frac{\pi}{4}$) is pivotal in securing optimal outcomes in a quantum game environment, it hints that a similar level of 'entanglement' could be instrumental in fostering collaborative synergy among various stakeholders, including researchers. Metaphorically, this implies that policies should be geared towards augmenting the degree of entanglement, thereby fostering transparency and trust.

In summary, the findings from the quantum game simulations advocate for policy development that focuses on increasing entanglement while concurrently enhancing transparency and building trust. Such strategies are poised to achieve Pareto optimality in cooperative scenarios, effectively surpassing the limitations inherent in conventional game theory approaches.

### Policy implications and recommendations

This article seeks to reinterpret the issues surrounding the Korean government's R&D investment, previously criticized as an 'R&D cartel', through a new lens: that of a strategic game. This reinterpretation is achieved by framing the interactions between the government (the investors) and researchers (the R&D implementers) within a quantum game model, conceptualized as a quantum information strategy model. The article posits that the Korean R&D investment environment can be modeled as a quantum game, featuring two primary agents:

**Table 5. Comparison of R&D cartel, R&D bureaucracy, and R&D monopoly.**

| | R&D Cartel | R&D Bureaucracy | R&D Monopoly |
|---|---|---|---|
| Definition | • *Agreements between competing companies to collaborate on R&D activities, often to control the pace of innovation and share markets.* | • *A system where R&D decisions and funding are controlled by a complex and inefficient administrative structure.* | • *A situation where one entity dominates R&D activities in a specific field or market.* |
| Main Features | • *Sharing of collaborative R&D costs and risks between competitors.*<br>• *Potential reduction in competition and possibility of market manipulation.* | • *Involves extensive procedural requirements and administrative controls.*<br>• *Slow decision-making process leading to inefficiencies.*<br>• *Often associated with government or large institutional structures.* | • *R&D capabilities and resources are concentrated in the hands of one entity.*<br>• *Leads to a lack of competition and innovation.*<br>• *Increased barriers to entry for other companies.* |
| Impact on Innovation | • *Positive aspects include collaboration and resource pooling.*<br>• *Negative aspects include reduced competition and potential stifling of innovation.* | • *Can be slow and limited due to procedural rigidity and lack of flexibility.* | • *Lack of competitive pressure to innovate.* |
| Market Dynamics | • *Induces coordinated market behavior which can limit competition but might accelerate certain innovations.* | • *Characterized by excessive regulation and control, potentially inhibiting market dynamics.* | • *Decreased market competition, increased prices, and limited consumer choice.* |
| Legal Considerations | • *Violation of antitrust laws if it leads to anti-competitive practices.* | • *Often legal, but prone to inefficiencies and lack of responsiveness to market needs.* | • *May be illegal if it results from anti-competitive practices.* |
| Representative Scholar | • *Harrington Jr, J. E. [38]*<br>• *Martin, Stephen. [39]* | • *Armstrong, J. S., & Green, K. C. [40]*<br>• *Damanpour, F.,& Evan, W. M. [41]* | • *Schumpeter, Joseph, A. [42]*<br>• *Tirole, Jean. [43]* |

Table 5 compares three organizational forms in the context of R&D: cartels, bureaucracies, and monopolies. It contrasts their definitions, main features, impacts on innovation, market dynamics, legal considerations, and representative scholars. The table suggests that while R&D cartels may foster cooperation, they risk market manipulation; bureaucracies may lead to inefficiencies due to procedural rigidity; and monopolies could stifle competition and innovation due to concentrated control.

the R&D manager (Alice) and the research performer (Bob). This model is akin to the classic prisoner's dilemma, emphasizing responsibility and sincerity.

In this model, Alice, representing R&D management, is inclined to focus on short-term or small-scale investments to mitigate risk. Concurrently, Bob, the research performer, tends to pursue short-term or small-scale tasks to maximize his utility, thus maintaining a Nash equilibrium. This equilibrium indicates a state where no participant can unilaterally improve their position without worsening the position of the other. As shown in Table 5, this study suggests that terms such as 'R&D bureaucracy' or 'R&D monopoly', which are often seen as moral hazards, may be more appropriate descriptions than the term 'R&D cartel'. The article underscores the potential of quantum game theory to extend and transform the classical game theory paradigm. This transformation is facilitated by introducing new strategic dimensions, such as quantum entanglement and superposition states. These dimensions offer fresh perspectives and policy alternatives for R&D innovation. Simulations within the quantum game model demonstrate the dynamic interplay of strategic configurations, highlighting how quantum entanglement can significantly alter the reward landscape, surpassing the traditional dilemmas inherent in classical models.

## Conclusion

### Summary of findings

A crucial aspect of this study is the illumination of how quantum strategies, particularly those employing entanglement, can facilitate Pareto optimal outcomes. In such scenarios, no player

can improve their payoff without adversely impacting other players. This quantum-based optimization, achieved through varying levels of entanglement, marks a significant departure from classical strategies, where mutual cooperation is often non-equilibrium and suboptimal. The study thus contributes valuable insights into the potential of quantum strategies to revolutionize traditional approaches to strategic decision-making in R&D investment.

This study offers a nuanced understanding that the efficacy of bilateral strategies in fostering cooperation and mitigating moral hazard is contingent not solely on the degree of entanglement but also on the dynamic fairness of the strategic (policy) response system, aligned with the fundamental compensation mechanism.

The simulation results for the aforementioned combinations of quantum strategies demonstrate that both players achieve Pareto optimality (3,3) with the configurations ('I', 'I'), ('$\sigma_x$', '$\sigma_y$'), ('$\sigma_y$', '$\sigma x$'), and ('$\sigma_z$', '$\sigma_z$'). The strategic implications and policy alternatives corresponding to these strategy combinations are elucidated as follows:

- **Strategy ('I', 'I')—Status Quo Maintenance**: This strategy represents a scenario where neither player alters their state, symbolizing stability and predictability in quantum games. It reflects a tendency towards preserving the status quo, with the sole change being the strength of entanglement. The corresponding policy alternative should involve implementing systems and policies that effectively evaluate research efforts. This can be achieved through enhancing system transparency, establishing a government investment technology notification system, and introducing a jury-type evaluation committee.

- **Strategy ('$\sigma_x$', '$\sigma_y$')—Complementary Change**: The '$\sigma_x$' and '$\sigma_y$' Pauli operators represent shifts in different dimensions. This combination signifies both players attempting strategic changes in diverse directions, fostering diversity and adaptability. The policy response should focus on defining clear responsibilities and obligations in R&D selection and performance. This involves establishing systems ensuring researchers' accountability and introducing a real-name R&D policy/planning system for R&D discovery and policy proposals.

- **Strategy ('$\sigma_y$', '$\sigma_x$')—Dynamic Response**: This combination, utilizing '$\sigma_x$' and '$\sigma_y$' operators in reverse order, implies a dynamic response strategy. The first player's '$\sigma_y$' operation influences the second player's '$\sigma_x$' strategy, reflecting a flexible response to changing environments. Policy alternatives should encourage long-term investment and prioritize innovative projects through a mandatory large-scale R&D quota system.

- **Strategy ('$\sigma_z$', '$\sigma_z$')—Cooperative Progress**: The selection of '$\sigma_z$' by both players indicates a joint progression towards a common goal. The '$\sigma_z$' operator maintains a specific state while inducing internal changes, signifying a cooperative and goal-oriented approach. As a policy alternative, there is a need to fortify cooperation within the R&D system, create a dynamic cooperative environment, and promote mutually beneficial strategies. This includes expanding rewards for research achievements, acknowledging sincerity in failures, and sharing accumulated information on R&D performance and planning management.

In conclusion, this study delineates the transformative influence of quantum strategy dynamics on game theory. It provides an integrated approach that merges the policy framework of quantum strategy and entanglement to advance innovation and collaboration in the research and development sphere.

## Limitations of the study and future research directions

In this study, policy alternatives derived from quantum game theory have been metaphorically applied to policy situations using the rotation axis and direction of the identity operator and

Pauli operators ('σx', 'σy', 'σz'). However, this methodology, while instrumental in translating the intricate concepts of quantum mechanics into a policy-making context, is not without significant limitations.

Firstly, there exists a discernible gap between abstraction and practical application. The operators and theoretical constructs of quantum mechanics are inherently abstract, and the direct correlation of these to tangible policy decisions may lead to discrepancies in practical application. Metaphorical interpretations and abstract models may not entirely encapsulate the complexities of real-world policy environments, potentially impacting the feasibility of the proposed policies.

Secondly, the approach is subject to the subjectivity inherent in metaphorical interpretation. The process of metaphorically applying the operator's axis and direction to policy alternatives introduces a degree of interpretation subjectivity. Different researchers may perceive the same quantum mechanical phenomenon differently, leading to varied interpretations and applications. This variability can raise concerns regarding the consistency and objectivity of the policy proposals.

Thirdly, the challenge of empirical verification presents itself. Policies based on quantum game theory are challenging to empirically validate. While integrating quantum mechanical concepts into social science contexts is intellectually stimulating, quantifiably measuring and verifying the efficacy of these approaches in real policy-making processes is a complex endeavor.

Fourthly, the applicability of policies derived from quantum game theory may be limited to specific contexts or situations. Not all policy issues are amenable to resolution within the quantum game theory framework. In certain cases, traditional game theory or other analytical tools may prove more pertinent.

By acknowledging these limitations and considerations, this study aims to provide a more comprehensive understanding of the applicability and practicality of quantum game theory, thereby establishing a groundwork for future research directions.

Lastly, in terms of practical use, it is the open use of data and information. This is also related to the construction of a quantum computing system according to technological development. In particular, it is an alternative to ensure policy transparency. Future quantum technologies, such as quantum communication, quantum sensing, and distributed quantum computation, rely on shared entanglement networks between spatially separated nodes. For a variety of parameters, our policy should improve previously known policies such as the "swap as soon as possible" policy with respect to latency and fidelity of end-to-end entanglement [44]. To obtain these results, the entanglement distribution must be modeled using a Markov decision process, and then reinforcement learning (RL) algorithms, etc., must be applied to discover the queue-replacement policy, thereby promoting cooperation between each temporally and spatially distant node. It will need to be quantified to support decision-making. To achieve this, the interaction between data providers, users, and regulators should be analyzed through quantum evolution game theory, and the need for open data for sustainable digital economy development should be emphasized [9]. For reference, the Python code used in this study is provided as S1–S3 Appendices.

## Supporting information

**S1 Appendix. Python code for calculations Table 4 and graphs Fig 3.**
(DOCX)

**S2 Appendix. Python code for graph Fig 4.**
(DOCX)

**S3 Appendix. Python code for graph** Fig 5.
(DOCX)

## Author Contributions

**Conceptualization:** BangRae Lee.

**Methodology:** Jongyeon Lim.

**Supervision:** BangRae Lee.

**Writing – original draft:** Dongkyu Won.

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
