## [Decision Letter · Decision Letter 0]

10 Mar 2024

PONE-D-24-05836Quantum Game Strategy Solution for R&D Cartel: Reorganizing Government R&D Investment Strategy in KoreaPLOS ONE

Dear Dr. Lee,

Thank you for submitting your manuscript to PLOS ONE. After careful consideration, we feel that it has merit but does not fully meet PLOS ONE’s publication criteria as it currently stands. Therefore, we invite you to submit a revised version of the manuscript that addresses the points raised during the review process.

We look forward to receiving your revised manuscript.

Kind regards,

Academic Editor

PLOS ONE

Journal Requirements:

"Development and Utilization of Innovation Strategy Analysis Models for the Science and Technology Industry (K-23-L03-C04)"

4. Please note that funding information should not appear in the Acknowledgments section or other areas of your manuscript. We will only publish funding information present in the Funding Statement section of the online submission form. Please remove any funding-related text from the manuscript. 

5. We note that your Data Availability Statement is currently as follows: "All relevant data are within the manuscript and its Supporting Information files."

Reviewers' comments:

Reviewer's Responses to Questions

**Comments to the Author**

1. Is the manuscript technically sound, and do the data support the conclusions?

Reviewer #1: Partly

Reviewer #2: Yes

2. Has the statistical analysis been performed appropriately and rigorously? 

Reviewer #1: Yes

Reviewer #2: Yes

3. Have the authors made all data underlying the findings in their manuscript fully available?

Reviewer #1: No

Reviewer #2: Yes

4. Is the manuscript presented in an intelligible fashion and written in standard English?

Reviewer #1: Yes

Reviewer #2: Yes

5. Review Comments to the Author

Reviewer #1: In this paper, the authors introduce a model of the interaction between the government (as an investor) and researchers (as R&D implementers) within the framework of a quantum information strategy, redefining the issues related to the Korean government's R&D investment. The application of quantum entanglement and superposition state theory enables the model to provide innovative policy measures and governance strategies for government R&D investment. In addition, by providing Pareto-optimal solutions in the newly developed quantum game model, the authors attempt to address the dilemmas inherent in the government's existing R&D investment policy. It is an interesting result and I believe the method is valid. However, I have some concerns as follows:

1. The abstract lacks a description of the results, and the authors should describe key findings or valuable conclusions in the abstract. In addition, the abstract should be a complete paragraph and should not be segmented.

2. The author’s description of game theory is not comprehensive enough. I suggest the author refer to the following papers (doi: 10.1109/TCSS.2023.3324087; 10.5109/4793672). Besides, recent research points out the fact that the study of evolutionary game theory has many applications in both the social and natural sciences, and there are many recent studies pointing this, which would be useful in the introduction to make this important point, such as 10.1109/TSMC.2023.3315963; 10.1063/5.0099697. This would further improve the introduction and the outline of possible future research.

3. There are some formatting mistakes in the paper. I would suggest the author proofread the paper meticulously and thoroughly. For example, the indentation of sections and subsections is inconsistent. Besides, each formula should be numbered accordingly.

4. The strategic combination between Alice and Bod on page 4 and page 5 is duplicated, and it is suggested that the authors delete one of them.

5. The figure is missing in the body of the text! Moreover, the resolution of the figures in the paper is so low that it is difficult to see the information inside the images, the author should increase the dpi of each figure and put them in the right place in the body of the text.

6. The results in the paper, including figures or tables, in addition to being cited in the body of the text, are also required to be analyzed accordingly, i.e., what phenomenon each figure or table shows and what conclusions can we draw from them.

7. Are the strategies of the individuals in this paper determined at the beginning and do not change in the evolutionary process? If not, why is there no simulation of how strategies change in the system?

8. The setting of sections and subsections in the paper is a bit confusing. It is suggested that the authors set up the model and simulation results in this paper as different sections, and then subdivide the subsections for the model and results.

9. There are only about five references in this paper from the last 5 years and most of the authors’ references are from before 2019, which are too obsolete. The references should cite more papers from the past five years.

10. Some references contain errors and inconsistent formatting. It is difficult to give credit to research if even such elementary aspects of the work are not error-free. The references should be made error-free and formatted in agreement with the journal guidelines. The authors should double-check to correct these errors.

Reviewer #2: This paper applies quantum game theory to reinterpret the issues surrounding the Korean government's R&D investment. By offering a Pareto optimal solution within the newly developed quantum game model, the authors seek to resolve dilemmas inherent in the government's existing R&D investment policies. This study offers novel strategic insights and interprets the complex dynamics inherent in R&D investment decisions. The manuscript seems interesting and insightful and the produced results also fulfill the motivation of the article in their own way. I recommend this submitted manuscript to seem suitable to go forward, though I want to keep some of the following points to the authors, which they might find useful and necessary to answer:

1. The use of symbols in the equations needs to be standardized, such as whether the same symbol is italicized or not, which needs to be consistent throughout the text. In addition, it is suggested to number the equations for easier understanding and reading.

2. It's not common to use " betrayal" (Page 4, Section 2.2). Instead, this field often uses "defection".

3. When calculating , 1++ i2 equal to 1? When calculating , 1 - i2 equal to 1? In other words, i2 equal to 0? The authors should explain.

4. The figures are not clear. It is suggested to improve their clarity.

5. Some references contain errors and inconsistent formatting. It is difficult to give credit to research if even such elementary aspects of the work are not error free. The references should be made error free and formatted in agreement with the journal guidelines.

6. PLOS authors have the option to publish the peer review history of their article (what does this mean?). If published, this will include your full peer review and any attached files.

Reviewer #1: No

Reviewer #2: No

---

## [Author Response · Author response to Decision Letter 0]

8 Jul 2024

The authors have carefully considered the comments from the reviewers and editor and made every effort to incorporate their feedback into the revised manuscript. I have uploaded the <Manuscript>, <Response to Reviewers>, and <Revised Manuscript with Track Changes> files, as well as five image files with improved resolution.

---

## [Decision Letter · Decision Letter 1]

23 Jul 2024

Quantum Game Strategy Solution for R&D Cartel: Reorganizing Government R&D Investment Strategy in Korea

PONE-D-24-05836R1

Dear Dr. Lee,

We’re pleased to inform you that your manuscript has been judged scientifically suitable for publication and will be formally accepted for publication once it meets all outstanding technical requirements.

Kind regards,

Minyu Feng, Ph.D.

Academic Editor

PLOS ONE

Additional Editor Comments (optional):

Based on the two experts' feedback and my own reading, I gladly recommend the publication of this work.

Reviewers' comments:

Reviewer's Responses to Questions

**Comments to the Author**

1. If the authors have adequately addressed your comments raised in a previous round of review and you feel that this manuscript is now acceptable for publication, you may indicate that here to bypass the “Comments to the Author” section, enter your conflict of interest statement in the “Confidential to Editor” section, and submit your "Accept" recommendation.

Reviewer #1: All comments have been addressed

Reviewer #2: All comments have been addressed

2. Is the manuscript technically sound, and do the data support the conclusions?

Reviewer #1: Yes

Reviewer #2: Yes

3. Has the statistical analysis been performed appropriately and rigorously? 

Reviewer #1: Yes

Reviewer #2: Yes

4. Have the authors made all data underlying the findings in their manuscript fully available?

Reviewer #1: Yes

Reviewer #2: Yes

5. Is the manuscript presented in an intelligible fashion and written in standard English?

Reviewer #1: Yes

Reviewer #2: Yes

6. Review Comments to the Author

Reviewer #1: The paper is well revised, I recommend the acceptance, but the Fig. 2 should be refined for a better display.

Reviewer #2: Since the authors have revised the manuscript in accordance with the comments, the quality of the manuscript has been greatly improved. I recommend a current version to accept this work.

7. PLOS authors have the option to publish the peer review history of their article (what does this mean?). If published, this will include your full peer review and any attached files.

Reviewer #1: No

Reviewer #2: **Yes: **Xiaopeng Li

---

## [Editor Report · Acceptance letter]

10 Sep 2024

PONE-D-24-05836R1 

PLOS ONE

Dear Dr. Lee, 

I'm pleased to inform you that your manuscript has been deemed suitable for publication in PLOS ONE. Congratulations! Your manuscript is now being handed over to our production team.

Kind regards, 

on behalf of

Dr. Minyu Feng 

Academic Editor

PLOS ONE